# Prevalence of antibiotic-resistant *Escherichia coli* isolates from healthy chicken droppings

**Ali M. Hussein**[ID][1]*, **Ali J. Muhialdin**[2], **Rahman K. Faraj**[ID][3]

1 Department of Medical Microbiology, Faculty of Science and Health, Koya University, Koya 44023, Kurdistan Region – F.R., Iraq, 2 Medical Laboratory Technology Department, Kalar Technical College, Garmian Polytechnic University, Kalar, Kurdistan Region, Iraq, 3 Department of Chemistry, College of Science, University of Garmian, Kalar, Kurdistan Region, Iraq

* ali.hussein@koyauniversity.org

## Abstract

### Background

The increasing emergence of antimicrobial-resistant (AMR) *Escherichia coli* in poultry represents significant public health concern due to the risk of zoonotic transmission via the food chain. This study investigates the prevalence and resistance patterns of *E. coli* isolated from healthy broiler and indigenous chickens in Kifri City, Kurdistan Region, Iraq.

### Methods

A total of 200 cloacal swab samples were collected from healthy chickens (100 broilers and 100 indigenous). Standard bacteriological methods were used for *E. coli* isolation, followed by antimicrobial susceptibility testing against ten antibiotics using the Kirby-Bauer disc diffusion method. Additionally, molecular detection of resistance genes was performed via PCR.

### Results

The overall isolation rate of *E. coli* was 60%. Broiler isolates exhibited significantly higher resistance rates, including 100% resistance to ciprofloxacin and enrofloxacin, and >90% resistance to amoxicillin, amoxicillin-clavulanate, norfloxacin, and nitrofurantoin. In contrast, indigenous chicken isolates showed lower resistance, with the highest rates seen for amoxicillin-clavulanate (90%) and nitrofurantoin (85%). PCR analysis identified the presence of key resistance genes including *blaTEM*, *qnrS*, and *sul1* among multidrug-resistant isolates. Statistically significant differences ($p < 0.05$) were observed in resistance profiles between broiler and indigenous groups.

### Conclusion

The high prevalence of multidrug-resistant *E. coli* in broiler chickens underscores the urgent need for stricter antibiotic stewardship in poultry farming. The findings

**Data availability statement:** "All relevant data are within the paper.".

**Funding:** The author(s) received no specific funding for this work.

**Competing interests:** No declarations from any of the authors.

support Sustainable Development Goals (SDGs) 3 (Good Health and Well-being), 12 (Responsible Consumption and Production), and 15 (Life on Land) by promoting antimicrobial surveillance and sustainable livestock management.

## Introduction

Antimicrobial resistance (AMR) has become a critical global health concern, compromising the effective treatment of infectious diseases and threatening food safety, public health, and environmental sustainability [1]. In the livestock sector, particularly in poultry farming, the overuse and misuse of antibiotics for growth promotion, prophylaxis, and metaphylaxis have significantly accelerated the development of resistant bacterial strains [2,3]. This selection pressure fosters the emergence and dissemination of multidrug-resistant (MDR) organisms, such as *Escherichia coli* (E. coli), which may act as reservoirs of resistance genes transmissible to humans via the food chain, direct contact, or environmental contamination [4,5].

*E. coli* is a Gram-negative, facultative anaerobic bacterium commonly found as a commensal organism in the intestines of animals and humans. Although most strains are harmless, pathogenic and MDR variants pose severe threats due to their ability to acquire and disseminate resistance genes through horizontal gene transfer mechanisms such as plasmids, integrons, and transposons [6]. Genes such as *blaTEM*, *qnr*, and *sul* families have been frequently associated with resistance to β-lactams, fluoroquinolones, and sulfonamides, respectively [7,8].

In developing regions, such as Iraq, regulatory oversight of antibiotic use in poultry farming remains limited, and antimicrobial agents are frequently administered without veterinary supervision. Consequently, broiler farms in such areas have become hotspots for MDR bacteria that can be transmitted to humans, contributing to the burden of AMR-related infections and reducing treatment options [9]. The local poultry sector, especially indigenous or scavenger chicken farming, presents a unique model for comparison, given its minimal antibiotic exposure and traditional management practices. Comparing resistance profiles between broiler and indigenous chickens offers valuable insight into how farming practices shape microbial ecology and resistance dissemination.

In alignment with the **One Health approach**, which integrates human, animal, and environmental health, this study investigates AMR in *E. coli* isolates from healthy chickens in Kifri City, Kurdistan Region. It contributes to the growing evidence base needed to guide public health interventions and antimicrobial stewardship in the poultry industry Fig 1.

### Aim of the Study

This study aimed to determine the prevalence and antimicrobial resistance patterns of *E. coli* isolated from the droppings of healthy broiler and indigenous chickens. Furthermore, it sought to identify the molecular basis of resistance through the detection of relevant resistance genes and to evaluate differences in resistance patterns using

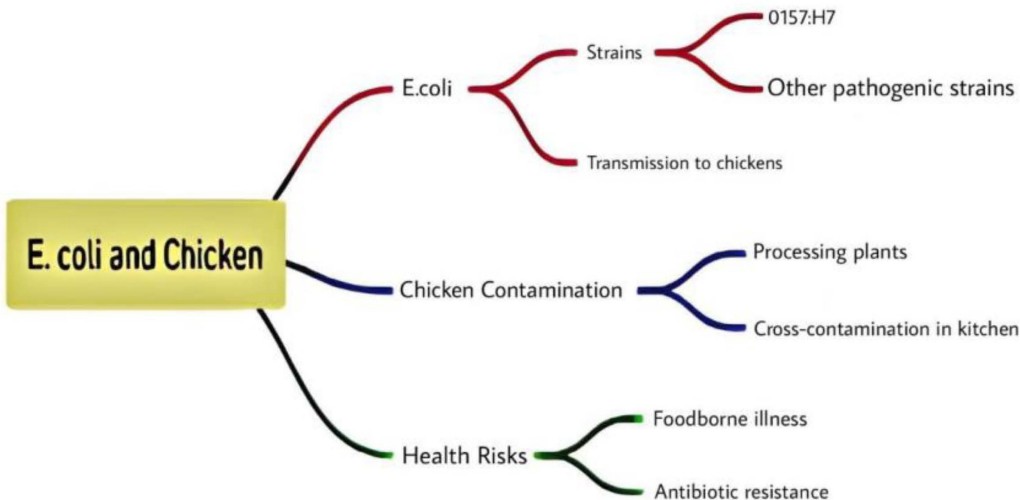

**Fig 1. Schematic overview of the interaction between *Escherichia coli* and chicken.** The figure illustrates the complex relationship between *E. coli* and poultry, highlighting major sources and transmission pathways. Commensal and pathogenic *E. coli* strains colonize the intestinal tract of chickens, from which they may disseminate to meat products during slaughter and processing. Contaminated poultry meat serves as a potential reservoir for zoonotic transmission to humans through improper handling, cross-contamination, or insufficient cooking. This diagram emphasizes the One Health perspective linking poultry production, microbial ecology, and public health.

statistical analysis. Ultimately, the study provides data to support antibiotic policy reform and sustainable poultry farming in line with SDGs 3, 12, and 15.

## Materials and methods

### Study design and sampling strategy

A cross-sectional study was conducted from December 2022 to April 2023 in Kifri City, Sulaymaniyah Governorate, Kurdistan Region, Iraq. The objective was to determine the prevalence and resistance profiles of Escherichia coli isolated from fecal samples of healthy chickens. A total of 200 cloacal swabs were collected from multiple poultry farms located in different areas within Kifri City to ensure broad geographic representation and minimize sampling bias, comprising 100 samples from broiler chickens reared under intensive farming systems and 100 from indigenous chickens raised in extensive/free-range conditions. The samples were aseptically transported in sterile containers on ice to the Microbiology Laboratory at the University of Garmian.

### Bacteriological isolation of *E. coli*

Swabs were pre-enriched in 9 mL of 0.1% buffered peptone water and incubated at 35°C for 24 hours. A loopful of enriched broth was streaked onto MacConkey agar, Eosin Methylene Blue (EMB) agar, and Xylose Lysine Deoxycholate (XLD) agar, followed by incubation at 37°C for 24 hours. Colonies showing characteristic morphology (e.g., metallic sheen on EMB) were sub-cultured and subjected to Gram staining and standard biochemical tests (Indole, Methyl Red, Voges–Proskauer, Citrate utilization).

### Antimicrobial Susceptibility Testing (AST)

Antibiotic susceptibility of confirmed *E. coli* isolates was evaluated using the Kirby-Bauer disc diffusion method on Mueller-Hinton agar, following CLSI guidelines [10]. The following antibiotic discs were tested as shown in the (Table 1).

**Table 1. The concentration of use Antibiotics.**

| Antibiotic | Abbreviation | Concentration |
|---|---|---|
| Ciprofloxacin | CIP | 10 µg |
| Cefotaxime | CTX | 30 µg |
| Amoxicillin-clavulanate | AMC | 20/10 µg |
| Ceftriaxone | CRO | 30 µg |
| Levofloxacin | LEV | 5 µg |
| Amikacin | AK | 10 µg |
| Amoxicillin | AMX | 10 µg |
| Norfloxacin | NOR | 30 µg |
| Nitrofurantoin | F | 100 µg |
| Enrofloxacin | ENR | 5 µg |

Zone diameters were measured and interpreted as Sensitive (S), Intermediate (I), or Resistant (R) according to CLSI breakpoints.

## Molecular detection of resistance genes

DNA extraction was performed on selected bacterial isolates using the boiling method. Briefly, bacterial pellets were suspended in sterile distilled water, heated at 95°C for 10 min, and centrifuged. The supernatant was used as the DNA template (Table 2).

Polymerase chain reaction (PCR) was conducted to detect resistance genes. The PCR reaction mixture (25 µL) contained 12.5 µL master mix, 1 µL of each primer (10 µM), 2 µL DNA template, and 8.5 µL nuclease-free water. Thermocycling conditions included an initial denaturation at 95°C for 5 min, followed by 35 cycles of denaturation (94°C, 30s), annealing (55–60°C, 30s), extension (72°C, 1 min), and a final extension at 72°C for 10 min.

## Statistical analysis

Data were analyzed using SPSS version 26. The chi-square test was used to compare the prevalence of antibiotic resistance between broiler and indigenous chicken isolates. A two-sample t-test was used to assess mean resistance levels across antibiotic groups. A p-value of <0.05 was considered statistically significant. Results are expressed as mean ± standard deviation or percentages where appropriate. Confidence intervals (95%) were reported for prevalence estimates.

## Ethics statement

This study involved non-invasive collection of cloacal swabs from live poultry during routine health checks. No anesthesia, euthanasia, or sacrifice was performed. The research protocol was reviewed and approved by the **Department of**

**Table 2. PCR primers used for resistance gene detection.**

| Gene | Resistance Type | Primer Sequence (5'–3') | Product Size (bp) | Reference |
|---|---|---|---|---|
| **blaTEM** | β-lactam | F: ATGAGTATTCAACATTTCCG<br>R: CTGACAGTTACCAATGCTTA | 516 | [7] |
| **qnrS** | Quinolone | F: GCAAGTTCATTGAACAGGGT<br>R: TCTAAACCGTCGAGTTCGGCG | 428 | [11] |
| **sul1** | Sulfonamide | F: CGGCGTGGGCTACCTGAACG<br>R: GCCGATCGCGTGAAGTTCCG | 433 | [12] |

PCR products were visualized by electrophoresis on a 1.5% agarose gel stained with ethidium bromide.

Biomedical Sciences, Cihan University–Erbil Institutional Animal Care and Use Committee (IACUC) under approval number **CUE-0101**, dated **23 December 2024**. Owner consent was obtained prior to sample collection.

## Results

### Prevalence of *E. coli* Isolates

Out of 200 cloacal swab samples analyzed, 120 isolates (60%) were confirmed as *Escherichia coli*. Among these, 60 isolates were from broiler chickens and 60 from indigenous (local) chickens. The prevalence of antibiotic resistance among isolates was calculated along with 95% confidence intervals to indicate the precision of the estimates. But *E. coli* did not significantly differ between the two groups (p = 0.12).

### Antimicrobial susceptibility profiles

The resistance profiles of *E. coli* isolates are summarized in Table 3. Broiler isolates exhibited notably higher resistance rates across all antibiotics tested. Resistance to ciprofloxacin and enrofloxacin was 100% in broilers, significantly higher than the 25% and 15%, respectively, in indigenous isolates (p < 0.001). High resistance was also observed in broiler isolates to amoxicillin (96.7%), amoxicillin-clavulanate (95.0%), norfloxacin (91.7%), and levofloxacin (95%). In contrast, indigenous chicken isolates showed comparatively lower resistance rates except for amoxicillin (85%) and amoxicillin-clavulanate (90%).

### Molecular detection of resistance genes

PCR analysis confirmed the presence of key resistance genes in *E. coli* isolatesAmong broiler isolates, 90% harbored blaTEM, 85% carried qnrS, and 70% possessed sul1. In contrast, indigenous isolates showed lower gene frequencies, with blaTEM detected in 50%, qnrS in 35%, and sul1 in 30% of isolates. The presence of these genes corresponded closely with phenotypic resistance to β-lactams, quinolones, and sulfonamides, respectively, indicating a strong association between molecular and phenotypic findings (Fig 2).

## Discussion

This study reveals a concerning prevalence of multidrug-resistant (MDR) *Escherichia coli* among fecal isolates from broiler chickens compared to indigenous poultry, highlighting a pressing threat to public health and food safety. The resistance rates among broiler isolates were substantially higher across nearly all tested antibiotics, which aligns with previous research from similar developing regions where antibiotics are routinely used as growth promoters and disease prophylactics in intensive poultry farming [2,13].

**Table 3. Antimicrobial susceptibility of *E. coli* isolates from broiler chickens (n = 60).**

| Antibiotic | Sensitive (%) | Intermediate (%) | Resistant (%) | p-value |
|---|---|---|---|---|
| Ciprofloxacin | 0 | 0 | 100 | <0.001 |
| Cefotaxime | 91.7 | 1.7 | 6.6 | 0.032 |
| Amoxicillin-clavulanate | 0 | 5.0 | 95.0 | <0.001 |
| Ceftriaxone | 81.7 | 5.0 | 13.3 | 0.048 |
| Levofloxacin | 1.7 | 3.3 | 95.0 | <0.001 |
| Amikacin | 16.7 | 33.3 | 50.0 | 0.003 |
| Amoxicillin | 0 | 3.3 | 96.7 | <0.001 |
| Norfloxacin | 3.3 | 5.0 | 91.7 | <0.001 |
| Nitrofurantoin | 1.7 | 6.6 | 91.7 | <0.001 |
| Enrofloxacin | 0 | 0 | 100 | <0.001 |

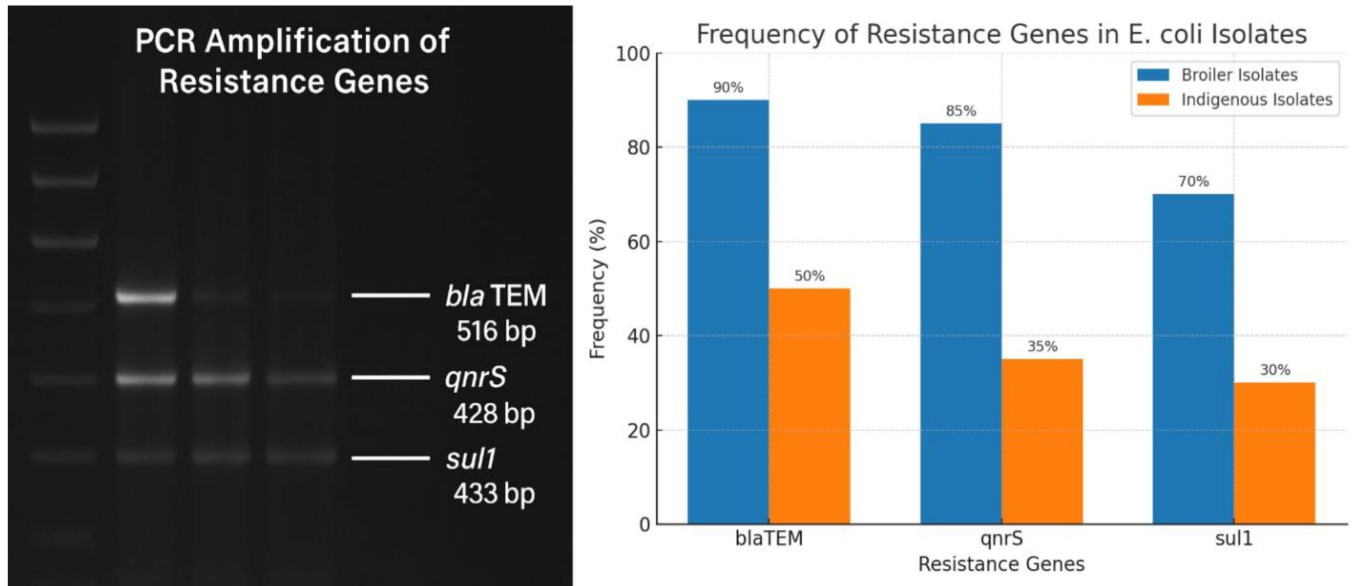

**Fig 2. Prevalence and distribution of antimicrobial resistance genes in *E. coli* isolates from broiler and indigenous sources.** The bar graph in this figure illustrates the frequency of detection for three antimicrobial resistance genes—*bla*TEM (516 bp), *qnrS* (428 bp), and *sull* (433 bp)—as determined by PCR amplification. Gene-specific amplicon sizes are indicated in base pairs (bp). Isolates are categorized by their origin: broiler isolates (n = 100) and indigenous isolates (n = 100). The numerical values atop the bars represent the precise percentage of positive isolates for each gene. The *bla*TEM gene demonstrated the highest overall prevalence and was significantly more abundant in broiler isolates (90%) compared to indigenous isolates (35%). In contrast, the frequencies of the *qnrS* and *sull* genes were comparatively lower and showed less disparity between the two isolate groups. This data highlights a distinct reservoir of β-lactam resistance (*bla*TEM) within the broiler-derived *E. coli* population.

The 100% resistance rate to fluoroquinolones (ciprofloxacin, enrofloxacin) in broiler chickens is particularly alarming, as these agents are classified as critically important antibiotics by the World Health Organization. The detection of *blaTEM*, *qnrS*, and *sul1* genes in these isolates corroborates the phenotypic findings and suggests that horizontal gene transfer via plasmids plays a vital role in disseminating AMR within poultry environments [7,6].

In contrast, indigenous chickens showed significantly lower resistance rates and reduced presence of resistance genes. This discrepancy may be attributed to the minimal or absent antibiotic exposure in free-range systems, where chickens are not routinely treated unless visibly ill. This supports previous observations that extensive, traditional poultry rearing methods may pose a lesser risk of AMR development [9,4].

The molecular detection of resistance genes in both poultry types illustrates the broader environmental and zoonotic implications. Genes like *qnrS*, associated with plasmid-mediated quinolone resistance, can facilitate rapid dissemination across bacterial species, increasing the risk of untreatable infections in humans [8]. The *blaTEM* gene, encoding β-lactamase, is frequently reported in extended-spectrum β-lactamase (ESBL)-producing *E. coli* strains, further exacerbating therapeutic challenges.

Statistical analysis confirmed significant differences (p < 0.001) in resistance profiles between broiler and indigenous isolates, underscoring the impact of farming practices and antibiotic exposure on resistance development. The high resistance rates in broilers also underscore the lack of regulatory enforcement in veterinary antibiotic use and the urgent need for stewardship programs.

This research has its boundaries. Due to the small sample size, the results may not be generalizable. Preferential sampling within particular Kifri City environments may lead to some area bias as well. Furthermore, the lack of molecular typing has

restricted the level of molecular characterization. It is suggested that follow-up studies with larger and more varied sample sizes, along with an exhaustive qualitative genomic approach, would help to substantiate and enrich the present findings.

## Public health implications and one health perspective

These findings underscore the critical need for integrated AMR surveillance strategies, improved veterinary oversight, and public awareness in rural and peri-urban poultry systems. AMR is not confined to clinical settings—it crosses the boundaries between humans, animals, and the environment. Healthy broiler chickens, often perceived as safe food sources, may act as silent reservoirs of MDR bacteria, contributing to the transmission of resistance through direct contact, manure usage, or contaminated meat products.

## Relevance to Sustainable Development Goals (SDGs) and sustainability

This study aligns directly with the **United Nations Sustainable Development Goals (SDGs)** by addressing the growing crisis of antimicrobial resistance (AMR) through research, awareness, and advocacy for sustainable poultry farming. The findings contribute to several key SDGs:

- **SDG 3 – Good Health and Well-being:** By identifying high levels of antibiotic-resistant *E. coli* in broiler chickens, the study highlights an urgent public health threat and the need for AMR surveillance. Promoting responsible antibiotic use in poultry production contributes to reducing zoonotic disease transmission and preserving the effectiveness of life-saving antibiotics.

- **SDG 12 – Responsible Consumption and Production:** The results advocate for more sustainable animal farming practices. Reducing the reliance on antibiotics in livestock production encourages ethical, eco-friendly poultry management that minimizes environmental contamination and long-term ecological risks.

- **SDG 15 – Life on Land:** Antibiotic misuse in agriculture affects biodiversity and soil microbiota through contaminated manure. The findings support the responsible integration of animal farming into ecosystems, preserving microbial and ecological balance.

- **One Health Integration:** The study embraces the One Health approach by acknowledging the interconnectedness of animal, human, and environmental health. It promotes interdisciplinary collaboration to combat AMR at all levels—from farms to hospitals.

By encouraging policymakers, veterinarians, farmers, and researchers to adopt sustainable practices and regulatory reforms, this research promotes a holistic path toward resilient food systems, healthy populations, and a safer environment.

## Author contributions

**Formal analysis:** Rahman K. Faraj.

**Investigation:** Ali J. Muhialdin.

**Methodology:** Ali J. Muhialdin.

**Software:** Rahman K. Faraj.

**Supervision:** Ali M. Hussein.

**Writing – review & editing:** Ali M. Hussein.

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
