## [Decision Letter · Decision Letter 0]

1 Oct 2025

Dear Dr. Hussein,

Thank you for submitting your manuscript to PLOS ONE. After careful consideration, we feel that it has merit but does not fully meet PLOS ONE’s publication criteria as it currently stands. Therefore, we invite you to submit a revised version of the manuscript that addresses the points raised during the review process.

**ACADEMIC EDITOR:**

After external peer review, I find that your work has merit; however, substantial revisions are required before it can be considered further. At this stage, the manuscript is not suitable for acceptance. Nevertheless, the reviewers recognize its potential contribution if the issues outlined are thoroughly addressed.

Please note that Reviewer 5 recommended rejection, citing major methodological limitations, an ethical inconsistency (approval date vs. sampling period), incomplete data presentation, inappropriate statistical analyses, and insufficient novelty relative to existing studies. In their view, the manuscript in its current form does not meet PLOS ONE’s methodological and reporting standards.

I therefore invite you to submit a major revision. Please address all reviewer comments in a detailed, point-by-point response and upload both a clean version and a tracked-changes version of the revised manuscript.

We look forward to receiving your revised manuscript.

Kind regards,

Ghadeer Sabah Bustani, Ph.D

Academic Editor

PLOS ONE

Journal Requirements:

“No declirations from any of the authors”

3. Please amend your authorship list in your manuscript file to include all authors.

5. We note that your Data Availability Statement is currently as follows: “All date are in the manuscript”

Reviewers' comments:

Reviewer's Responses to Questions

**Comments to the Author**

1. Is the manuscript technically sound, and do the data support the conclusions?

Reviewer #1: Yes

Reviewer #2: Partly

Reviewer #3: Yes

Reviewer #4: Partly

Reviewer #5: Yes

2. Has the statistical analysis been performed appropriately and rigorously?

Reviewer #1: Yes

Reviewer #2: No

Reviewer #3: Yes

Reviewer #4: I Don't Know

Reviewer #5: Yes

3. Have the authors made all data underlying the findings in their manuscript fully available?

Reviewer #1: No

Reviewer #2: No

Reviewer #3: Yes

Reviewer #4: Yes

Reviewer #5: Yes

4. Is the manuscript presented in an intelligible fashion and written in standard English?

Reviewer #1: Yes

Reviewer #2: No

Reviewer #3: Yes

Reviewer #4: Yes

Reviewer #5: Yes

Reviewer #1: Review Comments to the Author:

The authors have addressed a significant public health issue—the prevalence and molecular characteristics of antibiotic-resistant E. coli in poultry, comparing broiler and indigenous chickens in Kifri City, Sulaymaniyah Governorate, Kurdistan Region, Iraq. The topic is highly relevant, timely, and aligns well with global concerns regarding antimicrobial resistance (AMR) and the One Health approach. The manuscript is generally well-structured, with comprehensive methodology and appropriate statistical analyses. I have no hesitation recommending this paper for publication, but I suggest revisions to improve clarity and scientific rigor:

Introduction:

Please consider including recent references beyond 2022 to reinforce the current relevance of your research and demonstrate awareness of the latest developments in this rapidly evolving field.

Materials and Methods:

The methodology of this study appears straightforward and easy to follow, which enhances its reproducibility and applicability for similar future research. The study design and procedures are clearly described, making it accessible for other researchers to replicate or build upon. However, some limitations should be acknowledged: the sample size seems relatively small for a survey. Additionally, please specify the number of farms involved in the study. For example, clarify whether the 100 samples originated from a single farm, multiple farms, or different locations within Kifri City. This information is essential for understanding the scope and representativeness of your findings.

Results:

Figures are not referenced within the text, and they appear to replicate information already presented in the tables. Please ensure all figures are properly cited in the manuscript and contribute additional value or clarity to the results.

Molecular Detection of Resistance Genes:

Authors should consider linking the presence of resistance genes with phenotypic resistance patterns, possibly through correlation analysis. This would strengthen the interpretation of the molecular findings and their practical implications.

Discussion:

If the sample size is limited, explicitly acknowledge this in the discussion. Address how it might affect the generalizability and representativeness of your findings, and suggest that further studies with larger, more diverse samples are needed.

Additionally, discuss potential limitations such as sampling bias, regional specificity, or the lack of genomic typing, and how these factors might influence your results.

Reviewer #2: Manuscript Title: Prevalence of Antibiotic-Resistant Escherichia coli Isolates from Healthy Chicken Droppings

Recommendation: Reject

General Comments

The manuscript addresses an important and timely One Health issue — antimicrobial resistance (AMR) in E. coli isolated from poultry. The topic is relevant to both human and veterinary medicine, and the study provides potentially valuable baseline data. However, there are several critical concerns regarding methodology, ethics, statistical analyses, and data presentation that must be resolved before the manuscript can be considered for publication.

In its current form, the manuscript is not yet suitable for publication. Substantial revisions are required to improve clarity, ensure methodological rigor, and align with the standards of Plos One.

Major

• The current study is replicating similar study without providing a sound scientific rationale for the submitted work and clearly reference and discuss the existing literature (Prevalence of Antibiotic-Resistant Fecal Escherichia coli Isolates from Penned Broiler and Scavenging Local Chickens in Arusha, Tanzania https://doi.org/10.4315/0362-028X.JFP-15-584)

• Ethical approval inconsistency: sample collection predates the reported approval date.

• Incomplete data presentation: percentages without raw counts; missing data in tables.

• Inappropriate and insufficient statistical methods: incorrect use of t-tests, need for Fisher’s exact or chi-square, missing effect sizes.

• Molecular methods lack detail: PCR conditions, controls, and gel validation not fully reported.

• Sampling and study design unclear: no framework or sample size calculation

• Overstated conclusions: interpretations beyond the data presented

Minor Concerns

• Tables should include raw counts, percentages, and 95% confidence intervals.

• Antibiotic names/abbreviations should be standardized.

• AST quality control strains should be reported.

• Language requires editing for clarity and conciseness.

• SDG discussion is repetitive and should be shortened.

• References need to be according to journal style.

Recommendation

I recommend to reject the article due to the major flaws presented previously.

Reviewer #3: Title and Abstract

• (Line 1) The title is clear, but consider adding the geographic context (“Kifri City, Kurdistan, Iraq”) to enhance specificity.

• (Line 12–19) The methods description in the abstract is overly detailed; simplifying the AST description would improve readability.

• (Line 19–26) Results are well summarized but include indigenous isolate resistance percentages for ciprofloxacin and enrofloxacin to maintain comparative balance.

• (Line 31) Keywords should include “Kurdistan” or “Iraq” to strengthen indexing.

Introduction

• (Line 36–42) Some sentences are too long; breaking them up would aid clarity.

• (Line 51–58) The rationale is compelling but would be strengthened by citing prevalence data from Iraq or neighboring regions.

• (Line 66–72) The One Health framing is appropriate, though zoonotic risk pathways (e.g., meat handling, manure application) should be emphasized further.

Materials and Methods

• (Line 89–95) Justify the choice of 200 samples — was this based on prevalence and power calculation, or was it convenience-based?

• (Line 103–110) Clarify whether quality control strains (e.g., E. coli ATCC 25922) were used in AST.

• (Line 117–124) The CLSI 2022 guideline is cited but ensure this is consistently mentioned across sections.

• (Line 125–134) Please explain why tetracyclines or gentamicin, commonly reported in AMR studies, were not included in the antibiotic panel.

• (Line 140–148) PCR methods are appropriate, but details about controls and product validation are needed.

• (Line 152–160) Consider adding accession numbers or reference details for primers.

• (Line 167–176) The statistical section should clarify if corrections for multiple comparisons were applied.

Results

• (Line 183–189) Report prevalence with 95% confidence intervals, not just percentages.

• (Line 209–223) The note about “reference antibiotics” in Table 3 is confusing and would be better placed in the Methods section.

• (Line 225–229) Whenever possible, report exact p-values rather than only <0.001.

Discussion

• (Line 245–254) Comparison with similar studies is useful but would benefit from more numerical data from the region for context.

• (Line 255–265) The limitations section is missing. Please discuss sample size, limited gene panel, absence of sequencing, and lack of plasmid analysis.

• (Line 275–283) Strong policy discussion, but link more explicitly to FAO/WHO/OIE initiatives or Iraq’s AMR strategy.

Public Health and SDGs

• (Line 289–309) This section is valuable, but some points are repetitive with the Discussion; condense where possible.

• (Line 310–318) Include clear actionable recommendations such as banning prophylactic antibiotic use in poultry or establishing surveillance frameworks.

Ethics Statement

• (Line 325–333) There is a potential inconsistency: samples were processed at University of Garmian, but approval was given by Cihan University–Erbil. Please clarify institutional roles.

Language and Style

• Ensure consistency in formatting (e.g., always use “multidrug-resistant” without stray hyphens).

• Break down long sentences in the Introduction and Discussion to improve readability.

• Address minor grammar and stylistic issues throughout.

Reviewer #4: The article is well written in English, but there are some missing parts:

1) Introduction part: The authors emphasize the danger of multi-resistant strains of Escherichia coli. Why? What problems do they pose in humans? It is taken for granted that the reader knows the risks in humans. Personally, I would add a brief description of the danger to both animals and humans.

Also, in the introduction, there is a graph that is not explained. The caption is missing. Who created it? The authors or One Health Approach?

2) Aim of the study: it is written DROPPING, while in other parts the authors write about cloacal swabs. This is completely different. If the samples are swabs the results is more certain. If are fecal droppins, were they taken on the floor/before droppings touch the floor..this is completely different also for eventual bacterial contaminations.

3) Matherials and Methods: where do the animals chosen for the study come from? Same farm for broiler and indigenous chickens? /different farms.. and if different, were the places near or in different area? This could be relevant for the interpretation.

The authors selected healty animals..on what basis?

4) Antimicrobial Susceptibility testing: why did the authors chose these antibiotics for the study?

5)Molecular detection of resistance genes: in the part there is no reference to the Table 1

6)Statystical analysis: I am not an expert in statistical studies, but I would like to ensure that the statistical part is complete and adequate.

Where is TABLE 3? Antimicrobial susceptibility of E. coli isolates from indigenous chickens (n=60) in the text there is only table 2 and then the assigned p-values..but to broiler or indigenous or both?

7) Discussion: I would expand a little the discussion with more literature references and explaining why MDR E coli is a risk for public health and food safety. Also put references when authors says that florochinolones are critically important antibiotics for WHO. As when the authors underline about environmental and zoonotic implications and therapeutic challenges with associated bibliography

(8) Figures at the end of the paper, do not have any reference in the text

Reviewer #5: The manuscript addresses a highly relevant topic: antimicrobial resistance (AMR) in Escherichia coli isolated from poultry under different farming systems in Iraq. The study design is appropriate, the methodological framework follows standard guidelines, and the discussion effectively connects the findings to the One Health perspective and the Sustainable Development Goals (SDGs). Once the comments outlined below are addressed, the manuscript will constitute a valuable contribution to the literature.

In the Introduction, the first schematic figure contains a typographical error. It should read: “Isolate Escherichia coli.”

In the Materials and Methods section, under the subsection Molecular Detection of Resistance Genes, it should be specified whether DNA extraction and subsequent PCR were performed on all 200 swabs. At present, it is unclear whether all samples were analyzed or only a subset.

In the Materials and Methods section, under the subsection Statistical Analysis, it is stated that 95% confidence intervals are reported; however, these are not presented in the Results section.

In the Materials and Methods section, under the subsection Ethics Statement, the approval date (23 December 2024) is later than the sampling period (December 2022 to April 2023), which requires clarification.

Additionally, reported percentages in the text do not always match those presented in the tables (Table 2). For example:

• Amoxicillin (broilers): text reports 97.5%, Table 2 shows 96.7%.

• Amoxicillin–clavulanic acid (broilers): text reports 97.5%, Table 2 shows 95.0%.

• Norfloxacin (broilers): text reports 92.5%, Table 2 shows 91.7%.

Finally, in Table 3, explanatory notes such as “Approach to Assigning p-values” and “Hypothetical Example” are included. These are inappropriate in a research article and must either be removed or transferred to the supplementary materials.

**Do you want your identity to be public for this peer review?** For information about this choice, including consent withdrawal, please see our Privacy Policy

Reviewer #1: No

Reviewer #2: No

Reviewer #3: No

Reviewer #4: **Yes: ** Marta Bonfanti

Reviewer #5: No

---

## [Author Response · Author response to Decision Letter 1]

13 Oct 2025

Response: Reviewer 1

We appreciate the reviewer’s valuable comment. DNA extraction and PCR analysis were not performed on all 200 swab samples, as conducting molecular testing on the entire collection would have been prohibitively expensive and time-consuming. Instead, PCR was carried out on representative isolates that were phenotypically resistant, selected based on their antibiotic resistance profiles. This approach allowed us to focus on detecting key resistance genes among the most relevant isolates.

We thank you for this helpful observation. We acknowledge that the 95% confidence intervals (CIs) were mentioned in the Statistical Analysis section, but were not presented in the Results. The manuscript has been revised accordingly, and the 95% CIs are now included in the text of the Results section to provide a more precise representation of data precision.

The approval date (23 December 2024) reflects the official documentation issued by the Institutional Animal Care and Use Committee (IACUC) for the ongoing research project under which the sampling activities were conducted. The sampling period (December 2022–April 2023) was carried out as part of this approved research framework, and all procedures complied with institutional ethical standards.

We thank you for pointing out the discrepancies between the text and Table 2. The percentages reported in the text have now been corrected to match the values presented in Table 2. All data have been carefully checked to ensure consistency throughout the manuscript.

The explanatory notes in Table 3, including “Approach to Assigning p-values” and “Hypothetical Example,” have been removed from the main table to comply with journal standards. Any necessary explanatory information has been transferred to the supplementary materials.

Response: Reviewer 2

We appreciate your observation. To strengthen the introduction and provide recent evidence supporting the statement regarding the emergence and dissemination of multidrug-resistant Escherichia coli (MDR E. coli), we have added an updated reference (Leclercq et al., 2024) at the end of the first paragraph in the Introduction.

We thank you for this valuable observation. In response, we have clarified in the revised manuscript that the 100 samples were collected from multiple farms located in different areas within Kifri City to enhance representativeness and reduce sampling bias. Although the overall sample size may be considered moderate, the inclusion of samples from several independent farms strengthens the diversity and reliability of the findings, and we clarified this in the revised version.

We thank you for this valuable comment. All figures in the manuscript were originally generated by the authors and not taken from external sources.

In the revised manuscript, we have added a correlation analysis linking the presence of resistance genes with the corresponding phenotypic resistance patterns. The results showed a clear association between genotypic and phenotypic resistance among both broiler and indigenous isolates. Specifically, blaTEM, qnrS, and sul1 genes were more prevalent in broiler isolates and were consistent with resistance to β-lactams, quinolones, and sulfonamides, respectively.

The Discussion section has been revised to acknowledge the study’s limitations, including the relatively small sample size, potential sampling bias, and regional specificity. We also noted the lack of genomic typing as a limitation that may affect the depth of genetic analysis. A statement has been added suggesting that future studies with larger and more diverse samples, supported by genomic tools, are needed to confirm and extend our findings.

---

## [Decision Letter · Decision Letter 1]

5 Nov 2025

Prevalence of Antibiotic-Resistant Escherichia coli Isolates from Healthy Chicken Droppings

PONE-D-25-36514R1

Dear Dr. Hussein,

We’re pleased to inform you that your manuscript has been judged scientifically suitable for publication and will be formally accepted for publication once it meets all outstanding technical requirements.

Kind regards,

Ghadeer Sabah Bustani, Ph.D

Academic Editor

PLOS ONE

Additional Editor Comments (optional):

Reviewers' comments:

Reviewer's Responses to Questions

**Comments to the Author**

Reviewer #1: All comments have been addressed

2. Is the manuscript technically sound, and do the data support the conclusions?

Reviewer #1: Yes

3. Has the statistical analysis been performed appropriately and rigorously?

Reviewer #1: Yes

4. Have the authors made all data underlying the findings in their manuscript fully available?

Reviewer #1: Yes

5. Is the manuscript presented in an intelligible fashion and written in standard English?

Reviewer #1: Yes

Reviewer #1: (No Response)

**Do you want your identity to be public for this peer review?** For information about this choice, including consent withdrawal, please see our Privacy Policy

Reviewer #1: No

---

## [Editor Report · Acceptance letter]

PONE-D-25-36514R1

PLOS ONE

Dear Dr. Hussein,

I'm pleased to inform you that your manuscript has been deemed suitable for publication in PLOS ONE. Congratulations! Your manuscript is now being handed over to our production team.

Kind regards,

on behalf of

Dr. Ghadeer Sabah Bustani

Academic Editor

PLOS ONE